# HIV transmission dynamics and population-wide drug resistance in rural South Africa

Steven A. Kemp [1,2,10], Kimia Kamelian[1,10], Diego F. Cuadros [3,10], PANGEA Consortium*, Vukuzazi Team*, Mark T. K. Cheng [1], Elphas Okango[4], Willem Hanekom [4,5], Thumbi Ndung'u[4,5], Deenan Pillay[5], David Bonsall[2], Emily B. Wong [4], Frank Tanser[6], Mark J. Siedner [4,7,8,9] & Ravindra K. Gupta [1,4] ✉

Despite expanded antiretroviral therapy (ART) in South Africa, HIV-1 transmission persists. Integrase strand transfer inhibitors (INSTI) and long-acting injectables offer potential for superior viral suppression, but pre-existing drug resistance could threaten their effectiveness. In a community-based study in rural KwaZulu-Natal, prior to widespread INSTI usage, we enroled 18,025 individuals to characterise HIV-1 drug resistance and transmission networks to inform public health strategies. HIV testing and reflex viral load quantification were performed, with deep sequencing (20% variant threshold) used to detect resistance mutations. Phylogenetic and geospatial analyses characterised transmission clusters. One-third of participants were HIV-positive, with 21.7% having detectable viral loads; 62.1% of those with detectable viral loads were ART-naïve. Resistance to older reverse transcriptase (RT)-targeting drugs was found, but INSTI resistance remained low (<1%). Non-nucleoside reverse transcriptase inhibitor (NNRTI) resistance, particularly to rilpivirine (RPV) even in ART-naïve individuals, was concerning. Twenty percent of sequenced individuals belonged to transmission clusters, with geographic analysis highlighting higher clustering in peripheral and rural areas. Our findings suggest promise for INSTI-based strategies in this setting but underscore the need for RPV resistance screening before implementing long-acting cabotegravir (CAB) + RPV. The significant clustering emphasises the importance of geographically targeted interventions to effectively curb HIV-1 transmission.

South Africa remains at the epicentre of the global HIV-1 pandemic, with an estimated 7.5 million people, or 18.3% [15.6–20.5%] of the population, living with HIV-1 as of 2021[1]. Over the last two decades, there has been a reduction in the annual incidence of new infections, from 510,000 to ~210,000, resulting from an unprecedented public health effort to increase access to antiretroviral therapies (ART)[1]. This progress is noteworthy given the scale of the epidemic and highlights the effectiveness of concerted public health efforts. Nonetheless, HIV continues to pose significant morbidity and mortality risks, underscoring the need for ongoing vigilance and intervention[2].

[1]Department of Medicine, University of Cambridge, Cambridge, UK. [2]Pandemic Science Institute, Big Data Institute, University of Oxford, Oxford, UK. [3]Digital Epidemiology Laboratory, Digital Futures, University of Cincinnati, Cincinnati, OH, USA. [4]Africa Health Research Institute, KwaZulu-Natal, Durban, South Africa. [5]University College London, London, UK. [6]University of Stellenbosch, Cape Town, South Africa. [7]Division of Infectious Diseases, Massachusetts General Hospital, Boston, MA, USA. [8]University of KwaZulu-Natal, Durban, South Africa. [9]Harvard University, Cambridge, MA, England. [10]These authors contributed equally: Steven A. Kemp, Kimia Kamelian, Diego F. Cuadros. *Lists of authors and their affiliations appears at the end of the paper. ✉e-mail: rkg20@cam.ac.uk

A key challenge in the management and control of HIV-1 has been the selection and spread of drug resistance[3,4], with a noted global increase in non-nucleoside reverse transcriptase inhibitor (NNRTI) pre-treatment drug resistance (PDR)[5–7] prompting the World Health Organisation (WHO) to revise its first-line ART guidelines. This issue is particularly pronounced in sub-Saharan Africa, where NNRTI-based regimens have been compromised by systemic public health failures, especially in regions with sub-optimal viral load monitoring[6,8].

In response to these challenges, there has been a strategic pivot towards the use of integrase strand transfer inhibitors (INSTIs) such as dolutegravir (DTG) in first-line ART regimens[9]. This shift represents a critical evolution in ART strategy, but it also necessitates robust surveillance to monitor the emerging resistance to these newer antiretrovirals. Policy now recommends that new patients start DTG-based ART as well as ART-treated individuals with viral loads <1000 copies/mL. Although reports suggest a low prevalence of DTG resistance in ART-naïve individuals[10], there is a greater risk of DTG resistance amongst patients with detectable viral load at the time of the switch to DTG-based ART[11]. Previous resistance to lamivudine (3TC) or tenofovir (TDF) may adversely impact outcomes[12], with evidence showing PDR is associated with suboptimal responses to DTG-based regimens[13].

Understanding the degree and characteristics of population-level NNRTI resistance is of importance, given that long-acting (LA) CAB/rilpivirine (RPV) treatment is on the horizon[14], given the widespread use of NNRTIs in sub-Saharan Africa. Therefore, knowledge around the prevalence of NNRTI resistance-associated mutations relevant for RPV use in ART-naïve and ART-treated populations remains significant.

Another forthcoming LA-regimen combines the capsid maturation inhibitor lenacapavir (LEN) with the novel nucleoside reverse transcriptase translocation inhibitor (NRTTI) islatravir (EFdA). LEN resistance is readily selected in vitro, and the efficacy of EFdA is compromised by the rtM184V mutation. While there is substantial literature on rtM184V from high-income countries[15–17], updated data on the mutation's prevalence in recent cohorts from high-prevalence settings is critical for EFdA's potential use[18,19]. Yet, such empirical data on background resistance at the population level remain scarce in high-burden areas like South Africa. LA combinations are also poised to be used in prophylaxis and are predicted to be cost-effective, although their use is likely to increase the prevalence of drug resistance[20].

Despite advancements in ART, the HIV epidemic's landscape is heterogeneous. High HIV incidence rates persist in specific areas, like the uMkhanyakude district of KwaZulu-Natal (KZN), particularly among adolescent girls and young women[21]. This highlights the necessity for interventions tailored to the local epidemiology. Systemic challenges, including the shortage of community healthcare workers in KZN, have led to failures in care linkage, prompting the need for localised solutions[22,23]. The Vukuzazi programme, initiated in 2018, exemplifies such an approach, targeting HIV, tuberculosis, hypertension, and diabetes prevalence in the community while facilitating direct linkage to care[24].

The Vukuzazi study employed a cross-sectional survey approach to sample adolescent and adult residents aged 15 years or older within the Africa Health Research Institute demographic surveillance area in the uMkhanyakude district of KwaZulu-Natal, South Africa. This area is characteristic of rural South Africa, with a population predominantly of Black African descent, a 58% adult unemployment rate, and 66% access to piped water in their homes. The study was conducted over an 18-month period from May 25, 2018, to November 28, 2019. In terms of the recruitment process, individuals were initially contacted at their homes, identified using the geo-coordinates of their residence, and were invited to participate at mobile health camps that traversed the study area during the survey period. A substantial proportion of the eligible population was reached, with 26,460 individuals contacted out

of the 34,721 eligible (representing 76% of the eligible population). Out of these, 25,598 accepted the invitation to participate, and eventually, 17,118 individuals enroled in the study. This represents a 49% enrolment rate of the eligible population. The comprehensive reach of the study, combined with the methodology of using mobile health camps and home visits, ensured that the sampling was geographically representative of the demographic surveillance area. The use of inverse probability weights to account for non-response further aimed to mitigate potential bias and ensure that the prevalence estimates were representative of the population across different sex and age groups.

This study's objectives are twofold: (1) characterise antiretroviral resistance in this largely rural population via whole-genome HIV-1 sequencing and (2) map potential HIV transmission networks and their geospatial distribution.

## Results
### Population characteristics
The Vukuzazi study enroled 18,041 individuals, with 17,951 (99.6%) completing venepuncture for HIV testing (Fig. 1A, B). Among them, 6096 (33.9%) tested positive for HIV via ELISA testing. Notably, three participants had a positive HIV ELISA result but no viral load due to testing problems. The study's enrolment ratio was 2:1 female to male (assigned at birth) (Table 1). Approximately two-thirds (62.1%) of those with a positive HIV ELISA were ART-naïve at recruitment, based on the study's metadata and corroborated by electronic health record data. More than 85% of the known treatment regimens comprised of tenofovir/emtricitabine (FTC/TDF) and EFV ART. In those individuals testing positive for HIV, there was no relationship between viral load and time of enrolment into the study for ART-experienced and ART-naïve participants (Supplementary Fig. 1).

Of the 1232 venous blood samples with a viral load >40 copies/ml, 1097 resulted in successful whole-genome sequences. Post-QC, 47 sequences were excluded due to low or incomplete genome coverage. In total, 467 ART-naïve and 583 ART-experienced participants provided high genome coverage sequences for resistance and transmission analysis (Supplementary Fig. 2). Among the ART-experienced individuals, 27 individuals were on protease inhibitor (PI)-based second-line regimens, and 10 were on DTG-based ART.

### HIV-1 drug resistance
All drug-resistance-associated mutations were called at a minimum frequency of 5%. At this frequency, common drug resistance mutations included rtM184V, associated with 3TC and abacavir (ABC) resistance, observed in 32.6% of ART-treated and 1.9% of ART-naïve individuals (Fig. 2). Two TDF resistance mutations, rtK65R and rtrK70E were noted found in 12.0% and 6.2% of ART-experienced participants, respectively and in <1% of both ART-experienced and ART-naïve individuals. Thymidine Analogue Mutations (TAMs) including rtM41L, rtT215Y, rtD67N, rtK70R, rtT215F, and rtK219R/Q were found in <10% of ART-treated individuals and <2% of ART-naïve individuals. TAM is selected by zidovudine (ZDV) and stavudine (D4T), antiretrovirals that are no longer used as first-line ART. These findings, demonstrating the presence of TAMs in a large thymidine analogue-naïve population treated with first-line TDF-containing regimens, align well with studies across Sub-Saharan Africa evidencing thymidine resistance in similar regimens[25].

Among the detected NNRTI mutations, the prevalence in ART-experienced versus ART-naïve participants was as follows: rtK101E was found in 5.0% of ART-experienced and 1.5% of ART-naïve participants; rtK103N in 34.1% of ART-experienced and 9.4% of ART-naïve; rtV106M in 19.4% of ART-experienced and 2.8% of ART-naïve; and rtG190A in 8.2% of ART-experienced compared to 0.9% of ART-naïve participants. Importantly, the rtE138A mutation, known for conferring cross-resistance to RPV—a second-generation NNRTI used in LA injectable

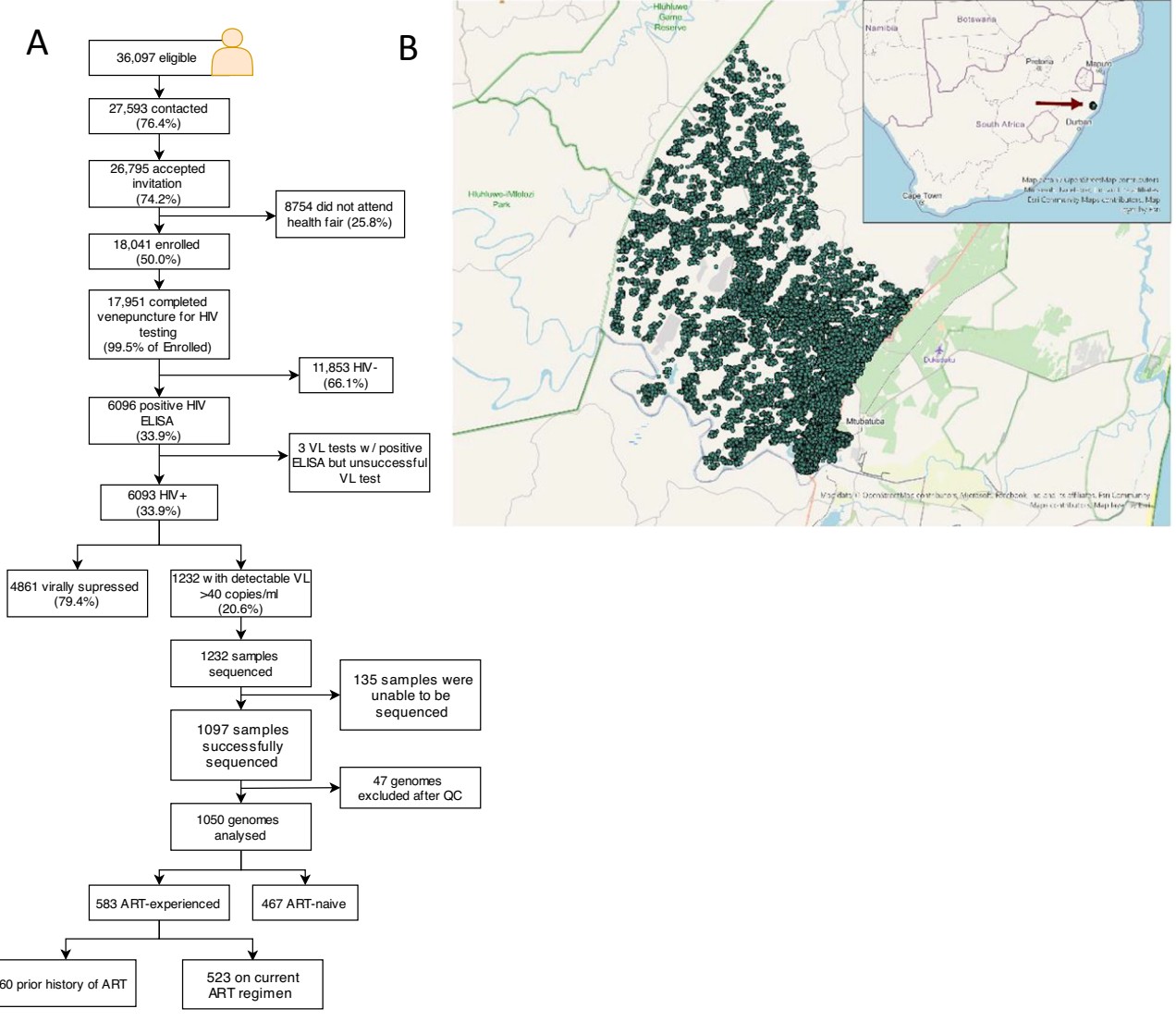

**Fig. 1 | Flow diagram of Vukuzazi cohort participation and location of study.**
**A** Individuals aged ≥15 years in the Africa Health Research Institute (AHRI) Demographic and Health Surveillance catchment area were eligible for the Vukuzazi study. From 1232 blood samples, 1050 genomes were successfully obtained from the cohort, and 36 were excluded due to poor coverage of *pol* and *env* genes. Downstream ART resistance and transmission analysis was conducted on 467 ART-naïve and 583 ART-experienced participants. **B** Map showing visualisation of the sampled individuals' location of residence in the Vukuzazi study, with dots representing the approximate locations within the study area. To protect participant confidentiality, a geographical random error has been introduced to each location, ensuring that the exact positions remain undisclosed. The underlying basemap is sourced from OpenStreetMap, © OpenStreetMap contributors.

ART alongside CAB, was observed in 6.5% of ART-experienced participants and 7.9% of ART-naïve participants (Figs. 2 and 3).

Resistance-associated mutations for INSTIs and PIs were rare in both ART-treated and ART-naïve individuals (Figs. 2 and 3). Both were found in fewer than 10 participants in the ART-treated and naïve groups. The most common INSTI mutations were inT97A and inE157Q (Fig. 2, both <1.5%). inT97A and inE157Q mutations are previously reported as polymorphic, although inT97A is associated with high-level resistance to DTG when in the presence of major INSTI mutations such as inG140S (https://hivdb.stanford.edu). It is important to note that this study was completed prior to the large-scale DTG rollout. Very few participants were on a DTG-containing regimen, but they were virally suppressed.

We also investigated the mutational abundance of virus quasispecies in plasma samples to identify minority drug resistance-associated variants. We classified the proportion of participants with a mutation at low thresholds of 5% and 10%, 20% (approximately the sensitivity of Sanger sequencing), and higher thresholds of 50% and 90% (Supplementary Figs. 3 and 4). Analysis of low-frequency NRTI variants (<20%) showed that rtK65R and rtK219R were detected at only 5% in ART-naïve individuals but at frequencies >90% in ART-experienced participants. rtM184V was only observed at >90% frequency in both ART-naïve and ART-experienced participants, though the proportion of ART-naïve participants with this mutation was significantly smaller (3% vs. 33%) than in ART-experienced participants. Other minority resistance-associated mutations in ART-naïve individuals included rtK103S and rtY181I, which only occurred at frequencies of <10%.

Finally, coreceptor usage was analysed using deep sequence data in the V3 loop region. The genotypic prediction model reported that the majority of viruses (93.8%) were predicted to use CCR5 as the co-receptor alongside CD4.

## HIV-1 infections demonstrating genetic linkage
We identified a total of 202 instances of strong genetic linkage between at least two or more participants within a cluster (Fig. 4A).

**Table 1 | Population characteristics, ART status, and regimens**

|  | All (n = 1050) | ART-Naïve (n = 467) | Art-experienced (n = 583) |
|---|---|---|---|
| **Age group (years)** | | | |
| 15–24 | 218 (20.8%) | 116 (24.8%) | 102 (17.5%) |
| 25–34 | 378 (36.0%) | 189 (40.5%) | 189 (32.4%) |
| 35–44 | 261(24.9%) | 99 (21.2%) | 162 (27.8%) |
| 45–54 | 147 (14.0%) | 101 (21.6%) | 46 (7.9%) |
| >55 | 74 (7.0%) | 45 (9.6%) | 29 (5.0%) |
| **Sex (assigned at birth)** | | | |
| Female | 693 (66.0%) | 298 (63.8%) | 395 (67.8%) |
| Male | 357 (34.0%) | 169 (36.2%) | 188 (32.8%) |
| **Currently on ART?** | | | |
| Yes |  | – | 603[a] |
| No |  | – | 34 |
| Unknown |  | – | 565 |
| **ART regimen** | | | |
| NRTI + NNRTI |  |  | 348 (89.5%) |
| TDF, FTC, EFV |  | – | 329 |
| TDF, EFV, 3TC |  | – | 5 |
| AZT, EFV, 3TC |  | – | 5 |
| ABC, EFV, 3TC |  | – | 1 |
| d4T, EFV, 3TC |  | – | 1 |
| TDF, FTC, EFZ |  | – | 1 |
| TDF, NVP, 3TC |  | – | 3 |
| AZT, NVP, 3TC |  | – | 2 |
| TDF, FTC, NVP |  | – | 1 |
| **NRTI + PI** |  | – | 27 (6.9%) |
| TDF, 3TC, LPV |  | – | 14 |
| AZT, 3TC, LPV |  | – | 9 |
| ABC, 3TC, LPV |  | – | 2 |
| TDF, FTC, LPV |  | – | 1 |
| TDF, ATV, LPV |  | – | 1 |
| **NRTI + INSTI** |  | – | 10 (2.6%) |
| TDF, 3TC, DTG |  | – | 10 |

Data from 1202 participants with associated metadata for whom blood samples were collected and sent to the University of Oxford for sequencing. Of those, 1050 were successfully sequenced and analysed.

*ABC* abacavir, *EFV* efavirenz, *FTC* emtricitabine, *TDF* tenofovir, *3TC* lamivudine, *LPV/r* lopinavir/ritonavir, *DTG* dolutegravir, *d4T* stavudine, *NVP* nevirapine, *INSTI* integrase strand transfer inhibitors, *NNRTI* non-nucleoside reverse transcriptase inhibitors, *NRTI* nucleoside/nucleotide reverse transcriptase inhibitors, *PI* protease inhibitors.

[a]Participant is currently taking ART, but regimens were unknown in 274 participants.

These 202 participants were divided into 86 high-confidence linked pairs or chains following pruning to avoid calling low-probability clusters. Among these, 63/86 (73.3%) were found to be transmission clusters involving just two individuals, forming a linked pair. The remaining clusters varied in size: 11 clusters (12.8%) each included three participants, 7 clusters (8%) included four participants each, one cluster (1.1%) consisted of five participants, and another single cluster (1.1%) comprised six participants. In total, 202 participants were linked to at-least one other individual in these 86 clusters. In the largest linked group comprising 6 participants (Fig. 4B), three had evidence of antiretroviral resistance (Fig. 4B), and three were wildtype (WT). In the second largest cluster of five participants, three were wildtype; one had only the NNRTI mutation rtE138A and the fifth had multiple NNRTI and NRTI mutations (Fig. 4C).

## Geospatial visualisations

The geospatial structure of epidemiological measures such as HIV viral load on treatment and antiretroviral resistance are graphically presented in Fig. 5. A varying level of HIV prevalence across the region with predominantly high HIV prevalence (>35%) is observed in the bottom right (southeast) portion of the sampled area (dark blue regions in Fig. 5A), whereas the prevalence of treatment failure does not exhibit a clear spatial pattern (Fig. 5B). Notably, the bivariate map delineating HIV prevalence and treatment failure identifies areas bearing a high burden of both measures (dark green in Fig. 5C), primarily concentrated in regions previously marked with high HIV prevalence. The geographical distribution of NNRTI (Fig. 5D) and NRTI (Fig. 5E) resistance prevalence reveals distinct patterns: NNRTI resistance is found primarily in the northern and southern surveillance regions, while NRTI resistance is more common in the northern region. The bivariate map presenting both mutations' prevalence identifies overlapping areas in the southern, central, and northern parts of the survey area (dark blue in Fig. 5F), as well as regions with high-NRTI-low-NNRTI prevalence (dark red) and low-NRTI-high-NNRTI prevalence (bright blue).

Phylogenetic linkages appear predominantly in the central and southern parts of the survey area showing high connectivity within these regions (Fig. 5G), largely coinciding with regions of high HIV prevalence. Highly connected nodes, ranging from 21 to 69 connections, are situated in the southern part of the survey area, where HIV prevalence is higher. Nodes with fewer connections (4–9) are mainly found in the central region of the survey area, where HIV prevalence is lower, as might be expected (Fig. 5).

## Discussion

ART guidelines currently recommend a combination of DTG, 3TC, and TDF as the first-line regimen for all eligible adults, adolescents, and children aged >10[26,27]. While the WHO has endorsed DTG-based ART as the preferred first-line regimen, understanding the patterns of NRTI and NNRTI antiretroviral resistance in both ART-naïve and ART-experienced patients remains crucial. This is particularly pertinent in the context of increasing evidence that PDR is associated with reduced efficacy and long-term failure of INSTI-containing first-line ART regimens[13]. Of note, however, resuppression in the context of drug resistance can occur in some individuals[28].

Currently, LA injectable treatments are centred around the combination of CAB/RPV. Reports suggest a 7.9% prevalence of RPV resistance mutations in ART-naïve populations, with a slightly higher 9.3% in ART-experienced population[29]. RPV mutations have been associated with virologic failure of LA CAB/RPV[30]. Similarly, we observed a concerning prevalence of RPV resistance at 6.5% in experienced participants and 7.9% in those untreated. These findings suggest that ART strategies involving a switch to LA CAB (INSTI) + RPV (NNRTI) may need to include prior screening for RPV resistance.

Regarding the future use of the novel RT inhibitor EFdA, the prevalence of rtM184V was significant in those with detectable viral load and on ART and present at lower thresholds in untreated individuals. rtM184V has been associated with reduced replication efficiency making it uncommon as a transmitted resistance mutation[17,25,31]. However, compensatory mutations such as rtL74I in the RT gene can restore viral fitness in the presence of rtM184V[19]. As such, the persistence of rtM184V poses a risk to the future use of LA combinational LEN and EFdA regimens.

rtK65R, a mutation that confers intermediate to high-level TFV and ABC resistance[18,32,33], was observed in 15% of ART-treated individuals with detectable viral load. This is higher than observed in high-income settings, although variation in prevalence based on subtype has been reported[34,35]. rtK65R and rtK219R were the only minority NRTI mutations (detected in <20% of the viral population) in ART-naïve individuals, reflecting findings from an earlier study in South Africa[32].

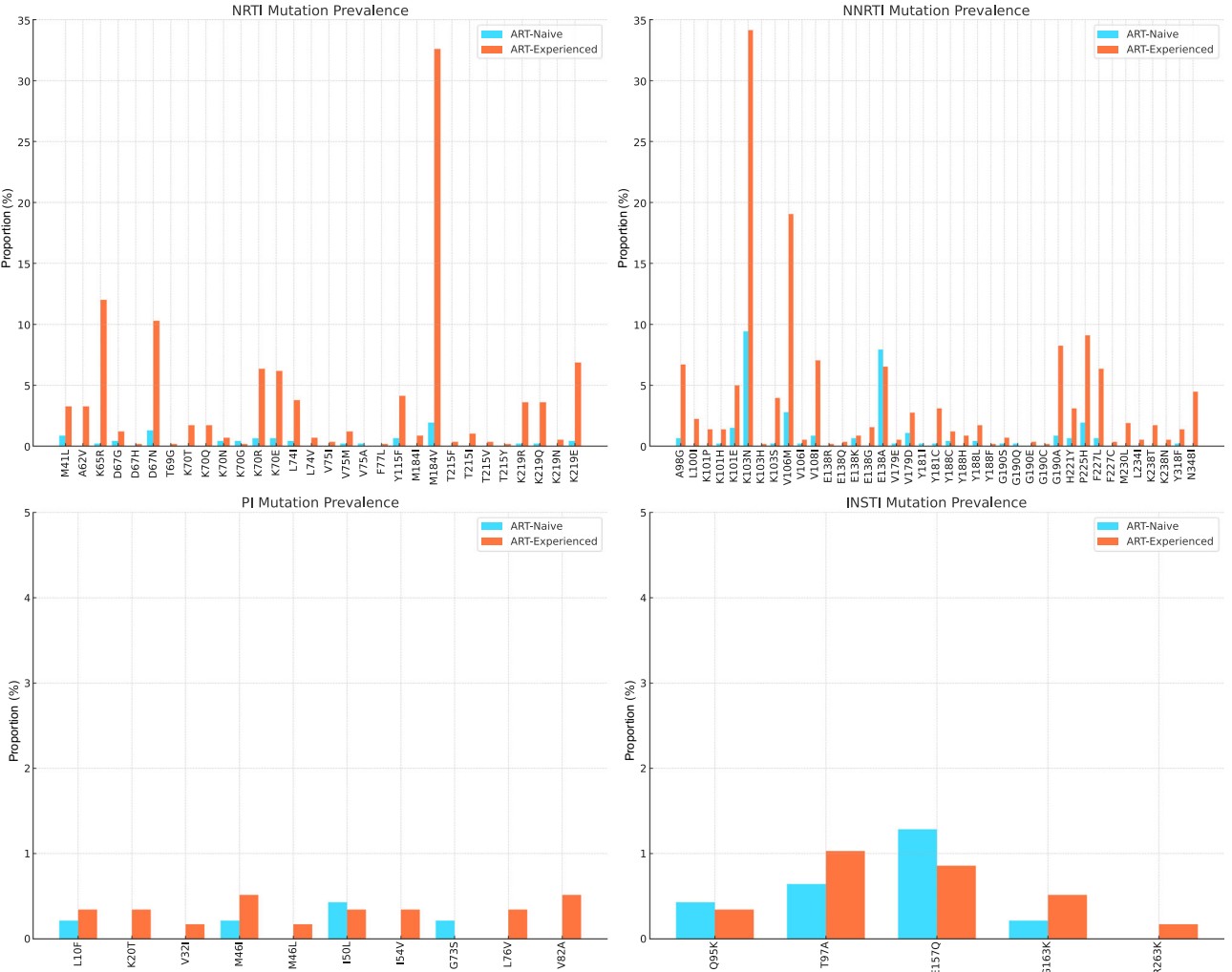

**Fig. 2 | Proportion of ART-naïve and ART-experienced participants with HIV-1 drug resistance-associated mutations at >5% variant abundance.** Mutations were determined by HIV-1 whole-genome sequencing using the Stanford University HIV Drug Resistance Database (v9.4) to determine resistance-associated mutations. Data are shown by drug class. NRTI nucleoside reverse transcriptase inhibitor, NNRTI non-nucleoside reverse transcriptase inhibitor, INSTI integrase strand transfer inhibitor, PI protease inhibitor.

This trend is likely due to the known propensity of subtype C to acquire these mutations spontaneously[36–39], attributable to differential pausing of RT at positions 64-66[40].

Before the scale-up of the INSTI-based ART[41], significant resistance to NRTIs and NNRTIs was identified, but resistance to PIs and INSTIs was minimal, as expected (Table 2). However, the low-level appearance of INSTI mutations within virus quasi-species was detected, including inR263K (DTG-resistance-associated mutation), among other secondary antiviral resistance mutations[42,43]. The inT97A mutation, a secondary drug-resistance-associated mutation[44], was also observed. inT97A can greatly enhance resistance in the presence of other INSTI mutations[45].

At present, resistance to integrase inhibitors in treatment naïve individuals is very low, with some individuals selecting major INSTI mutations[10,46]; however, ~10% of previously treatment-experienced individuals who go on to develop viremia whilst on an integrase inhibitor harbour INSTI mutations. Ongoing surveillance for INSTI resistance in the DTG treatment era is critical, given the highly promising results from CAB PreP studies[47,48] and emerging data showing that the use of LA CAB/RPV during acute HIV infection can result in the selection of major resistance mutations to INSTIs[49].

The application of next-generation sequencing for full-length HIV-1 genomes allowed us to conduct clustering analysis, revealing 86 high-confidence clusters, predominantly involving sequences without antiretroviral resistance. Our findings suggest transmission may occur before the initiation of ART. Geospatial analyses indicated clustering was most observable in areas with the highest HIV infection prevalence. Drug resistance to NNRTI - but not NRTI -coincided with HIV-1 prevalence geospatially, possibly due to the significant presence of transmitted NNRTI resistance in this population. Conversely, both NRTI and NNRTI resistance were geospatially associated with treatment failure or detectable viral load, reflecting a high probability of detecting NRTI and NNRTI resistance following treatment failure[7].

Building on previous study methodologies[50], our geospatial data visualisation provided a nuanced view of HIV prevalence, treatment failure, and drug resistance distribution. Despite the scale-up of ART, high HIV infection burdens and considerable levels of treatment failure were evident, signalling ongoing transmission and potential antiretroviral resistance emergence. This information could be vital for crafting effective public health strategies by pinpointing resource allocation for testing, treatment, and prevention efforts.

We used geospatial data visualisation to depict the application and purpose of the geospatial techniques utilised, particularly emphasising the illustrative nature of the methods involved. While our approach integrates methodologies such as Gaussian kernel

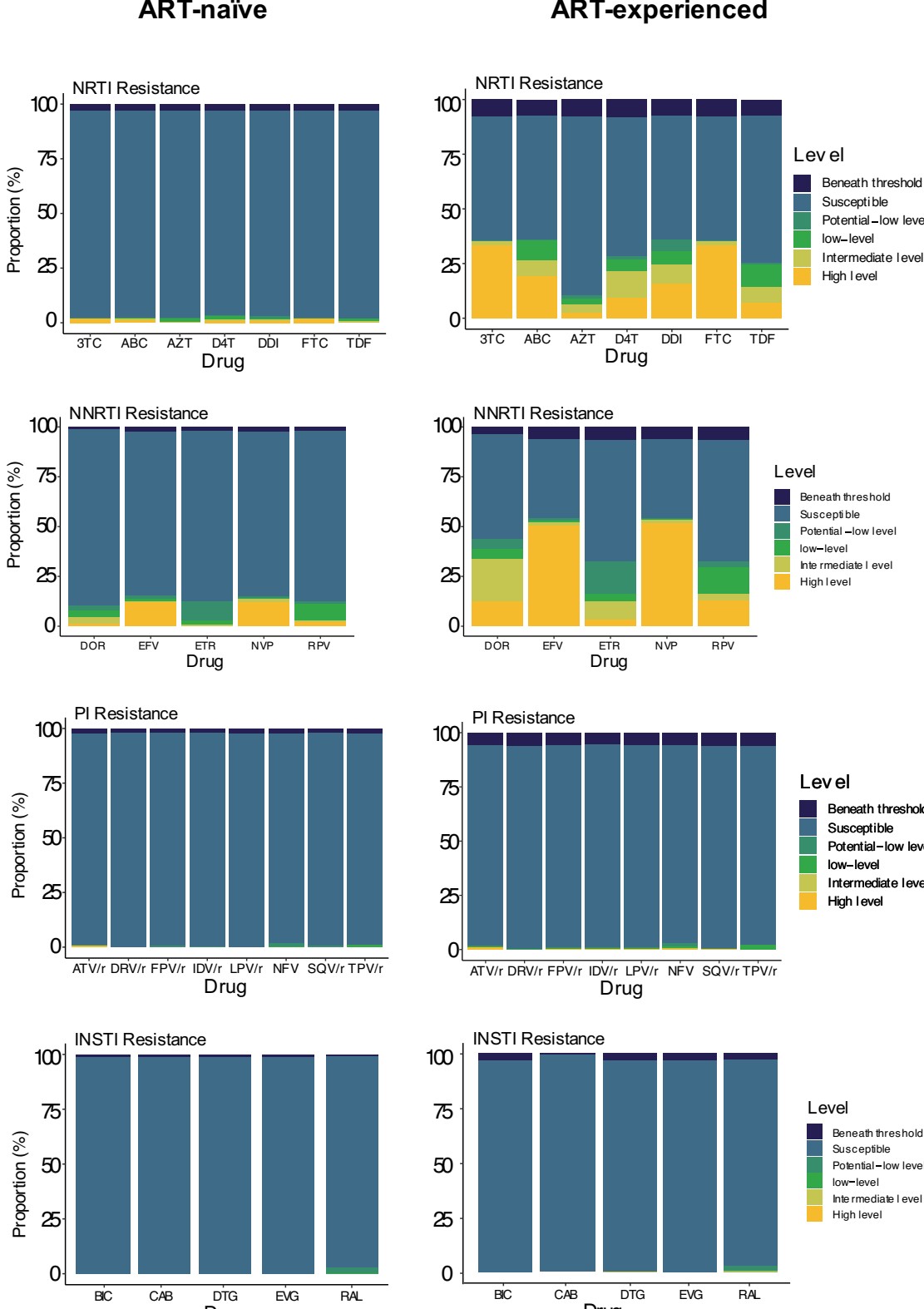

**Fig. 3 | Susceptibility to ART among ART-naïve (left) and ART-experienced (right) participants.** Beneath threshold: less than half of the sites had sufficient data at the stipulated threshold to determine resistance. Susceptible: all relevant sites had read at the stipulated threshold, and no mutations were detected. Data for four drug classes are shown. NRTI nucleoside reverse transcriptase inhibitor, NNRTI non-nucleoside reverse transcriptase inhibitor, INSTI integrase strand transfer inhibitor, PI protease inhibitor.

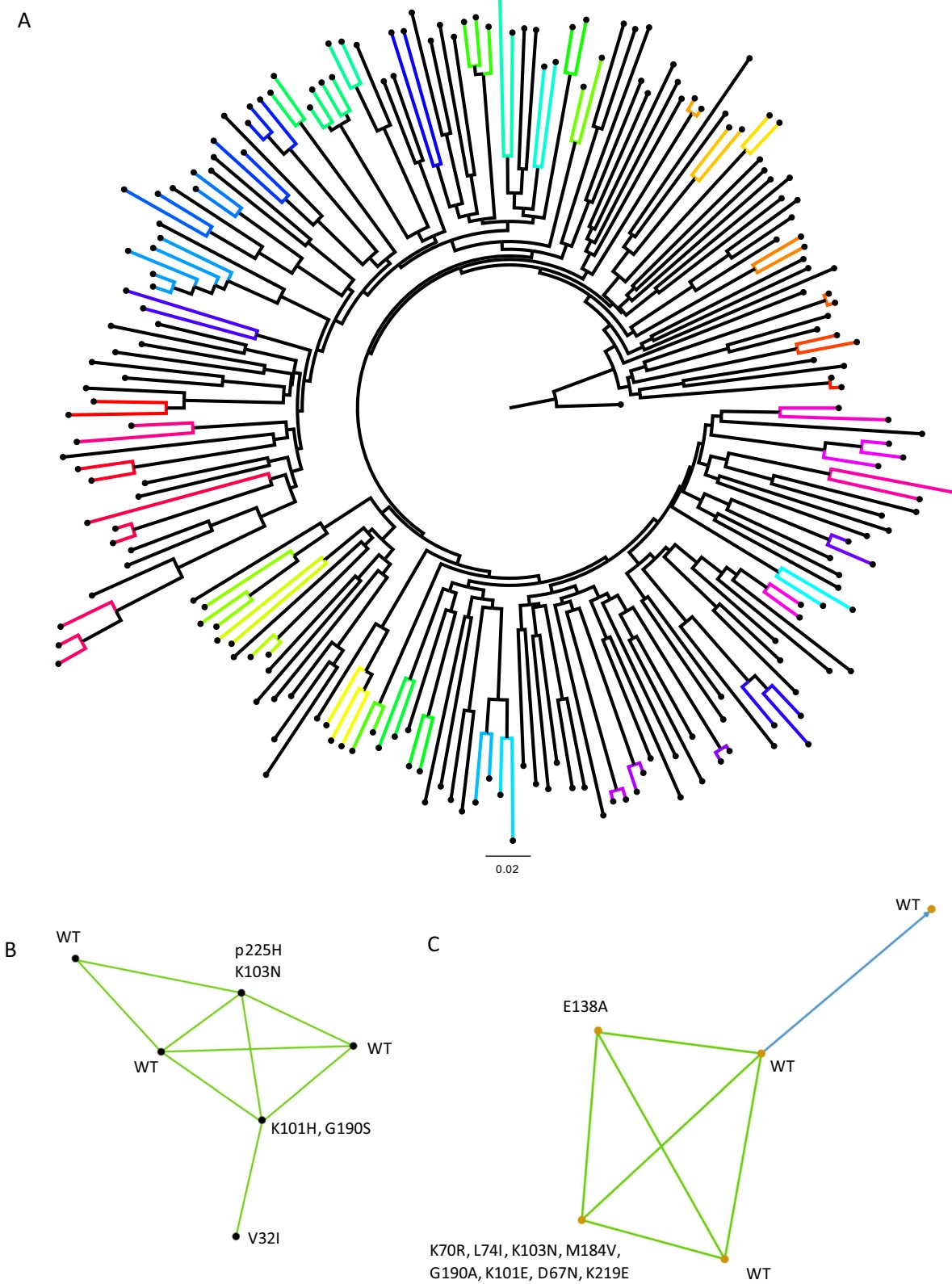

**Fig. 4 | Phylogenetic analysis of HIV-1 sequences and identification of transmission clusters. A** Maximum-likelihood phylogenetic tree of HIV-1 sequences that passed QC. Bootstrap support indicated at nodes. Clusters as defined by Cluster Picker using genetic distance threshold of 4.5 and statistical support by bootstrapping of >98% are coloured. **B** Linkage and resistance data on largest cluster with 6 participants. **C** Linkage and resistance data on the second largest cluster with 5 participants. Blue lines indicate >95% confidence and green indicate >80% confidence.

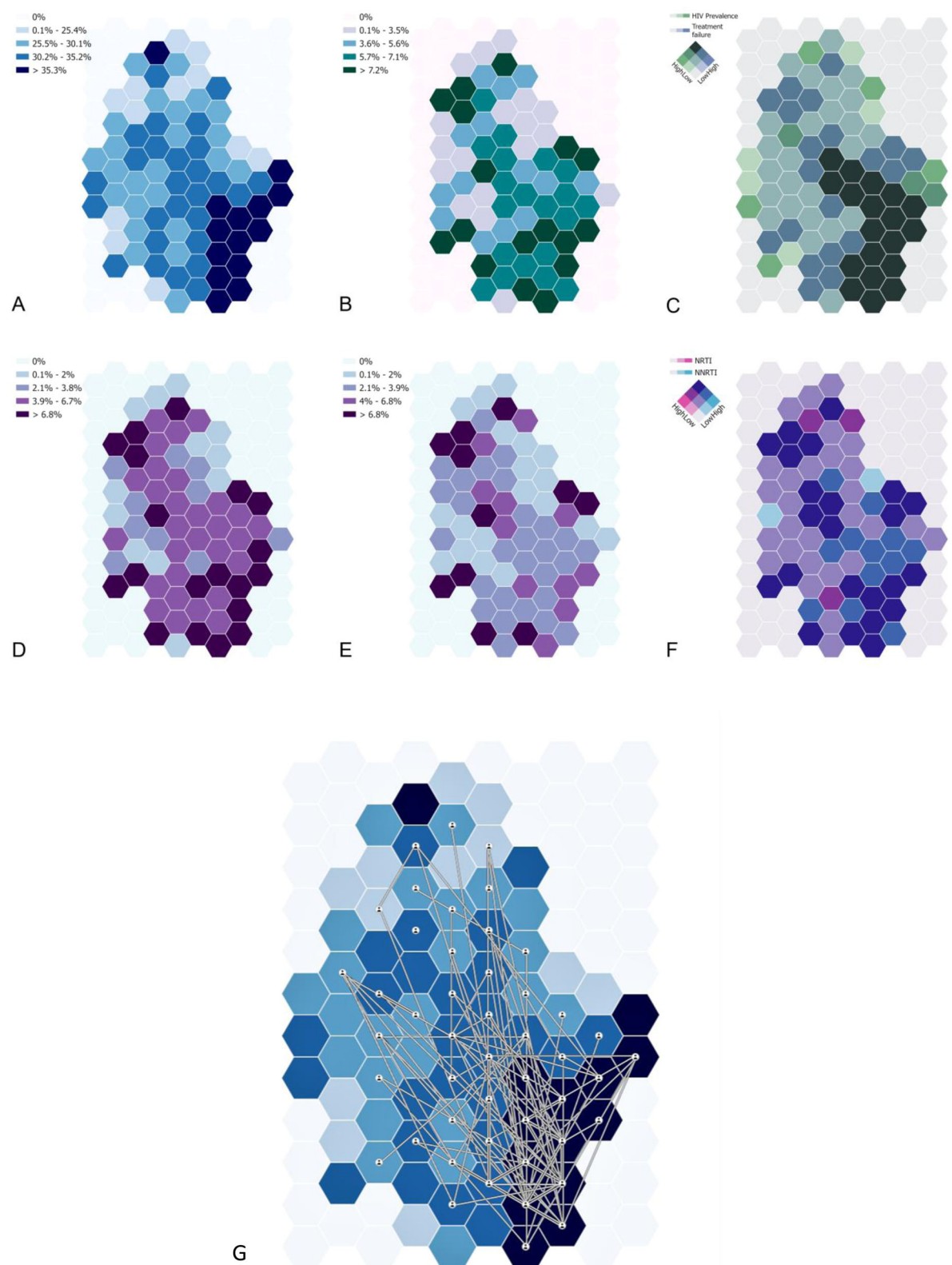

**Fig. 5 | Geospatial analysis of NRTI and NNRTI resistance in the uMkhanyakude District. A** HIV prevalence. **B** prevalence of treatment failure among HIV-positive individuals. **C** Bivariate map among HIV prevalence and treatment failure. **D** NRTI prevalence among HIV-positive individuals. **E** NNRTI prevalence among HIV-positive individuals. **F** bivariate map among NNRT and NNRTI. **G** High confidence phylogenetically linked participants plotted on the background of HIV prevalence.

interpolation, it is important to clarify that these techniques primarily serve to enhance the visual representation and comprehensibility of the spatial distribution of key variables, including HIV prevalence and resistance mutation prevalences. The principal intent behind these visualisations is to provide an intuitive and accessible interpretation of the spatial dimensions of our data rather than to conduct in-depth geospatial analyses. This distinction is crucial as it aligns with the overarching aim of the study to present data in a manner that

**Table 2 | Multi-class drug resistance and cases of protease inhibitor (PI) and INSTI resistance amongst all participants**

| ID | ART class | | | | No. of classes | ART status |
|----|------|-------|-----|-------|----------------|------------|
|    | NRTI | NNRTI | PI  | INSTI |                |            |
| 1  | –    | –     | –   | Yes   | 1              | ART-Experienced |
| 2  | –    | –     | Yes | –     | 1              | ART-Experienced |
| 3  | –    | –     | Yes | –     | 1              | ART-Experienced |
| 4  | –    | –     | Yes | –     | 1              | ART-Experienced |
| 5  | –    | Yes   | Yes | –     | 2              | ART-Experienced |
| 6  | –    | Yes   | Yes | –     | 2              | ART-Experienced |
| 7  | –    | Yes   | Yes | –     | 2              | ART-Experienced |
| 8  | Yes  | Yes   | Yes | –     | 3              | ART-Experienced |
| 9  | Yes  | Yes   | Yes | –     | 3              | ART-Experienced |
| 10 | Yes  | Yes   | Yes | –     | 3              | ART-Experienced |
| 11 | –    | –     | Yes | –     | 1              | ART-Naïve |
| 12 | –    | –     | Yes | –     | 1              | ART-Naïve |
| 13 | –    | –     | Yes | –     | 1              | ART-Naïve |
| 14 | –    | –     | Yes | –     | 1              | ART-Naïve |
| 15 | –    | Yes   | Yes | –     | 2              | ART-Naïve |

Three individuals exhibited three-class resistance. Seven showed PI resistance only. None of these participants existed in a linkage cluster.
*INSTI* integrase strand transfer inhibitors, *NNRTI* non-nucleoside reverse transcriptase inhibitors, *NRTI* nucleoside/nucleotide reverse transcriptase inhibitors, *PI* protease inhibitors.

supports, rather than directly contributes to, the inferential statistical findings.

High-resolution mapping of HIV prevalence, treatment failure, and antiretroviral-specific resistance offers insights into the potential for decentralised sampling in surveillance programmes. Regions marked by high HIV prevalence and ART failure require a robust response, including expanded testing initiatives. Mobile testing units and community health workers could be strategically placed in these high-need areas, potentially increasing diagnosis rates and linking individuals to treatment services more effectively. The maps can also guide adherence support programmes to regions where treatment failure is prevalent. By integrating different strategies, such as digital adherence tools, peer support groups, and community-based interventions, healthcare systems can aim to enhance patient outcomes and mitigate the burden of treatment failure[51]. This geospatial data can empower healthcare providers to optimise treatment by selecting ART combinations less susceptible to resistance.

The spatial dynamics of HIV transmission provide insight into the epidemic's spread. Identifying highly connected zones allows for targeted prevention efforts, such as increased condom distribution, education campaigns, and PrEP for high-risk individuals[52]. Understanding these interlinkages enables interventions to be more precisely aimed at community locations central to disease spread[50,53]. The high geospatial resolution transmission linkages identified, coupled with the distribution of antiretroviral-specific mutations, can inform contemporary local prescribing decisions and policy, particularly as we progress towards LA injectables. Thus, we advocate for ART programmes to be coupled with viral load and drug resistance monitoring using viral sequencing in order to enable clustering analyses to be scaled up[54].

The observed geospatial patterns of HIV are influenced by various factors, including mobility and migratory patterns[55]. Although study participants were effectively linked to healthcare services, the potential for HIV transmission during travel or migration persists, leading to decentralised transmission patterns. Social and sexual network dynamics are equally crucial, as HIV transmission is largely influenced by these structures, often extending beyond geographical limits. The distinct spatial distribution of NNRTI and NRTI resistance observed in the study is likely influenced by several factors, such as the historical deployment and use duration of these antiretrovirals within the community[56–58].

The Vukuzazi study's unique population-based approach, as opposed to clinic-based or smaller cohort studies, provided a comprehensive and less biased snapshot of the prevalence of current HIV resistance. Sequencing a large number of community samples offers a detailed picture of the HIV resistance landscape, critical for designing interventions and tracking transmission networks, offering a detailed perspective that is often lost in smaller, more selective sampling methods. This large-scale analysis not only enables more precise detection of prevalent resistance mutations but also facilitates more accurate inference and spatial geolocation of transmission networks within the community. Such granularity in data is rare and instrumental in shaping effective public health strategies and interventions.

Limitations of this study include its cross-sectional nature and capturing resistance at a single time point rather than longitudinally during an infection course[59]. Ideally, serial surveys would be undertaken to provide information on dynamics, particularly in the era of INSTI-based first-line ART. Furthermore, the data on treatment history required collation from more than one source and, in some cases, was incomplete. Finally, the small sample sizes in some locations made it difficult to draw robust conclusions regarding the geographic patterns of resistance across all parts of the study area.

In conclusion, our study delivers novel insights into the patterns of HIV drug resistance and linkages within a rural area with high HIV prevalence. HIV-1 whole-genome sequencing has facilitated not only the identification of linked infections but also a more intricate understanding of the HIV-1 drug resistance landscape at the population scale.

## Methods
### Study setting and recruitment
The Vukuzazi study recruited 18,025 participants (adolescents and adults aged ≥15) from their homes in the uMkhanyakude district, KwaZulu-Natal, South Africa, to healthcare screening for hypertension, diabetes, HIV, and tuberculosis. Full details of the Vukuzazi study methods and results have been previously reported[24,60]. Recruitment occurred between May 25, 2018, and November 28, 2019, during the Vukuzazi cross-sectional survey.

## Ethics

Ethical approval was obtained from the Ethics Committees of the University of KwaZulu-Natal (BE560/17), London School of Hygiene & Tropical Medicine (#14722), the Partners Institutional Review Board (2018P001802) to conduct the research in KwaZulu-Natal, Durban, South Africa in a single location. All participants provided written informed consent for HIV testing and ensuing analysis. Written informed consent was also obtained from legal guardians/representatives for participants under the age of 18.

## Data and blood sample collection

Mobile health clinics across the study area facilitated data collection. Research nurses compiled participants' medical histories, including prior HIV, tuberculosis, hypertension, and diabetes diagnoses. Blood samples from 17,949 participants were collected for HIV testing using the Genscreen Ultra HIV Ag-Ab enzyme immunoassay [Bio-Rad]. The HIV-1 RNA viral load was subsequently measured for immunoassay-positive samples, resulting in 6093 positive tests. Samples with detectable viral load defined as >40 copies/ml ($n = 1323$) underwent HIV-1 whole-genome sequencing.

## Whole-genome sequencing and bioinformatics

We employed the veSEQ-HIV[61] method for whole HIV-1 genome sequencing on the Illumina MiSeq platform following established protocols. We then used the bioinformatics pipeline, drmSEQ, to identify genotypic resistance, aligning codons with a database of 142 HIV reference sequences. We used the Stanford HIV Drug Resistance Database classification system, with 'high-level resistance' as our threshold for resistance. Drug resistance mutations were called at a minimum of 10 reads and 2% frequency. We obtained the participants' ART status (ART-experienced, $n = 583$; ART-naïve, $n = 467$) from the PANGEA consortium, TIER.net, and the Vukuzazi cohort study. HIV-1 subtyping occurred using SNAPPy v1.0. Prediction of co-receptor usage was made using TROPHIX (prediction of HIV-1 tropism). Available at: http://sourceforge.net/projects/trophix/).

## Transmission cluster identification and validation

All sequence data that passed quality control (QC) ($n = 1050$) was incorporated into a maximum-likelihood phylogeny, inferred using IQ-TREE v2.2.2 (1000 ultrafast bootstraps and a GTR + F + R6 model). For the initial identification of potential transmission clusters, we utilised ClusterPicker (v1.2.5) using an initial and main threshold of 98, a genetic distance of 4.5%, and a large cluster threshold of 10. This produced 171 potential clusters. To identify clusters with a high probability of containing accurate transmission chains, we utilised a backward stepwise logistic regression model testing various interactions between collection date intervals and patristic distances between sequence pairs[62] (Supplementary Fig. 6). The logit model allowed for refinement of clusters previously identified.

To determine transmission chains within clusters, we used Phylocsanner v1.8.2[63] using sliding windows of 150 bp across the entire HIV-1 genome. Phylogenies were constructed using IQTREE v2.2.2 and then analysed with the analyse_trees.R package.

## Geospatial analyses

We executed geospatial visualisation to describe the spatial distribution of several epidemiological parameters. We introduced a geographical random error to uphold participant confidentiality. We evaluated the prevalence of several epidemiological parameters by generating continuous surface maps, utilising a standard Gaussian kernel interpolation method[64]. Maps were created without geographical references, and a grid consisting of 108 hexagonal cells covering the surveillance area was used to aggregate the spatially interpolated estimates. The hexagonal grid cells, integral to the spatial structure of our visualisation, each have an area of 7.77 km². This

dimension was chosen to balance the need for detailed spatial resolution with the practical considerations of visual clarity and data confidentiality. It is important to clarify the role and application of these hexagonal grid cells within the context of our geospatial analysis and visualisation framework. The kernel interpolations, a central component of our geospatial methodology, were conducted using the actual point data derived from the study, allowing us to explore and represent the underlying spatial patterns of variables such as HIV prevalence and treatment failure prevalence. Following the generation of these continuous surfaces through kernel interpolation, the hexagonal grid cells were employed to group and visualise these interpolations and the HIV transmission linkages. This approach allowed us to present the geospatial data in a manner that is both accessible and informative, facilitating an intuitive understanding of the spatial distribution and intensity of the study variables across the study area.

The grid was further employed to illustrate HIV prevalence, treatment failure prevalence, NNRTI mutations, and nucleoside/nucleotide reverse transcriptase inhibitor (NRTI) mutations. Bivariate maps were generated to identify regions with overlapping epidemiological measures, such as high HIV prevalence coinciding with high rates of treatment failure or significant drug resistance mutations. We geospatially mapped the linkages of viral transmission among these individuals that were estimated by the previous phylogenetic analyses using the grid to aggregate the locations of these links into the centroids of each of the 108 cells of the grid that represented the nodes of the transmission links. We used the software ArcGIS Pro 3.1 (ESRI: ArcGIS Pro.x. Redlands, CA, USA: ESRI. 2020.) to construct the grid and generate the spatial data visualisations included in the study.

The Kernel Interpolation with Barriers tool in ArcGIS allowed us to estimate continuous surfaces for HIV prevalence, treatment failure prevalence, and NNRTI and NRTI mutation prevalence. We opted for the Gaussian kernel function, renowned for its bell-shaped curve and effectiveness in smoothing over various types of spatial data. This function was helpful in ensuring seamless transitions across the study area, thus maintaining the integrity of the spatial patterns observed in our data. A critical component of our methodology was the selection of the bandwidth or radius for the kernel function, which was set at 3 km. This parameter was calibrated based on previous studies that employed similar data in the same study population, ensuring that our spatial patterns are both representative and contextually relevant. Lastly, we utilised the prediction surface type, a choice that aligns with our objective to accurately represent the continuous surfaces of HIV-related measures across the study area.

The geographical random error was introduced by carefully displacing the geographical coordinates of each participant's location within a specified maximum distance. This displacement strategy was designed to ensure that the jittered locations remain reasonably close to the true locations, effectively obscuring the exact identification of participants' locations while preserving the overall spatial patterns pertinent to the findings of the study. The magnitude of jittering was determined, taking into account the size of the overall study area and the size of each grid hexagon used in the spatial analyses. Although the exact distances used for jittering cannot be disclosed due to confidentiality protocols, we ensure that the applied jittering sufficiently maintains the spatial integrity of the data. This ensures that the visualised patterns in our maps accurately represent the geographical distribution of the study variables without compromising individual privacy.

It is crucial to emphasise the delineation between the application of jittering for visualisation and the handling of data for the actual spatial analyses. The jittering of data points was strictly limited to the data utilised for visualisation purposes, such as maps included in publications or public databases. This measure is a standard in studies involving sensitive health data to prevent the potential identification of

individual participants. In contrast, the true, unaltered geographical coordinates were exclusively used for the spatial analyses that underpin our main findings. This approach guarantees that the statistical inferences and conclusions are based on the most accurate and unmodified data, thereby upholding the validity and reliability of the results of the study.

### Reporting summary

Further information on research design is available in the Nature Portfolio Reporting Summary linked to this article.

## Data availability

Sequencing data for the entire Vukuzazi/PANGEA cohort are available for download from GenBank, Accession: PRJEB19239 ID: 369369. No new sequencing was performed for this study. Datasets analysed during the current study are provided in the paper. Source data are provided with this paper, though viral loads have been redacted as they are potentially identifiable data.

## Code availability

Custom Python script relevant to the production of figures in the manuscript can be accessed at https://github.com/SteveKemp/Vukuzazi_manuscript.

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

## Acknowledgements

Wellcome Trust [Grant number 201433/Z/16/A], Bill & Melinda Gates Foundation, the South African Department of Science and Innovation, the South African Medical Research Council, and the South African Population Research Infrastructure Network. Wellcome Senior Fellowship to R.K.G. (WT108082AIA). M.J.S. receives additional funding from the US National Institutes of Health (K24 HL166024).

## Author contributions

Study conception, design, and administration: R.K.G., E.W., M.S., Vukuzazi study team; data collection: Vukuzazi study team, data analysis: S.A.K., K.K., D.C., M.C., E.O., M.S., F.T., R.K.G.; data interpretation: S.A.K., K.K., D.C., M.C., E.O., M.S., F.T., R.K.G.; manuscript preparation; S.K.

wrote the first draft of the manuscript, which was subsequently revised by K.K., D.C., M.C., E.O., M.S., W.H., T.N., D.P., D.B., E.W., F.T. and R.K.G. All authors reviewed the results and approved the final version of the manuscript.

## Competing interests

R.K.G. has received honoraria from ViiV and Gilead for advisory board participation. The remaining authors declare no competing interests.

## Additional information

## PANGEA Consortium

David Bonsall[2], Thumbi Ndung'u[4,5], Deenan Pillay[5], Ravindra K. Gupta ◉[1,4] ✉ & Mark J. Siedner ◉[4,7,8,9]

## Vukuzazi Team

Deenan Pillay[5], Willem Hanekom ◉[4,5], Emily B. Wong ◉[4], Mark J. Siedner ◉[4,7,8,9] & Thumbi Ndung'u[4,5]

A full list of members and their affiliations appears in the Supplementary Information.

