## [Peer Review File · Nature Communications]

HIV transmission dynamics and population-wide drug resistance in rural South AfricaREVIEWER COMMENTS

Reviewer #1 (Remarks to the Author):

In this manuscript, the authors use data from a large community-based cohort study conducted between 2018-2019 in the uMkhanyakude district of KwaZulu-Natal to explore patterns of drug resistance and transmission patterns. The manuscript is clearly written and leverages a robust data set to provide interesting insights into these topics in the study setting and period. I was asked by the editors to specifically provide comments regarding the geospatial analysis and defer to other reviewers with respect to the remaining analyses.

Comments:

1. Regarding the geospatial methods (lines 161-177): the authors use a relatively simple Gaussian kernel interpolation method in ArcGIS to estimate continuous surfaces of HIV prevalence, treatment failure prevalence, and NNTRI and NRTI mutation prevalence. These measures are then displayed on a hexagonal grid and combined with other analyses to provide insights into the spatial patterns of HIV epidemiology in the study region. In general, this approach seems reasonable. I have a few questions for clarification and comments regarding the methods:

1a. Lines 120-121: the authors refer appropriately to previous publications (ref 22-23) that have described the Vukuzazi study protocol. I might suggest that they add a brief sentence or two describing the sampling methodology – specifically, so that readers can understand whether the sampling is likely to be geographically representative or not without having to access to the previous manuscripts.

1b. Throughout the paragraph beginning with line 160, the authors refer to “geospatial data visualization”. I might suggest that they refer to “Geospatial analyses”, since some of their methods (e.g. Gaussian kernel interpolation”) go beyond simple visualization of the available geospatial data. This is a minor point.

1c. Line 165: Reference 25 (for the Gaussian kernel interpolation method) refers broadly to Waller & Gotway’s “Applied spatial statistics for public health data”. Would it be possible to give a more specific reference to the method as used in ArcGIS? For instance, was this done with the “kernel interpolation with barriers” tool, or some other tool? Similarly, in line 163-165: did the authors set any parameters for this tool (e.g. the bandwidth)? If so, it would be helpful for those to be specified to improve reproducibility and help the reader understand how much smoothing was applied.

1d. Lines 162-163: The authors added geographical random error to protect participant confidentiality, which is an important and common practice. How was the geographical random error introduced (e.g. what sorts of distances were used to displace the points)? Knowing how much jittering is applied (specifically, in the context of the size of the overall study area and the size of each grid hexagon) will help the reader understand the likely impact of this jittering in the analysis.

1e. Lines 165-167: Can the authors provide some idea of the size of the hexagonal grid cells? It’s a bit hard in the manuscript to understand the scale of the geospatial analysis, and this would help contextualize the analysis.

2. Figure 5: The authors show a series of hexagonal grid squares representing the uMkhanyakude district. For those readers who are less familiar with the district (such as myself), it would be helpful to provide some sort of geographical context in this figure – the grid alone is challenging to interpret. I might suggest, for instance, that the authors provide an inset map of the location of uMkhanyakude district within South Africa, and then also a map that overlays the hexagonal grid on a street map (or some other map that will help readers to understand the geography of the district and how it relates to the analytic grid that the authors have selected).

3. If possible, it would be helpful to see two additional figures to supplement the geospatial analysis. Both of these could be supplemental figures, but would help to provide more insight into the data and results:

3a. First, a map of the geospatial data coverage would be helpful – i.e. a simple map of the data collection points (after jittering) overlaid on the grid squares, or some other similar summary. This would help the readers to understand whether data collection was concentrated in certain areas of the grid, or if the distribution of collected data was more spatially diffuse.

3b. Second, it would be helpful to see some measure of uncertainty for the results of the kernel interpolation analysis. I presume that uncertainty differs between the different indicators (and across the geospatial grid) depending on data availability and density. I believe that ArcGIS is capable of producing standard error surfaces for some of its interpolation results, so – for instance – the authors could produce gridded maps of standard error for their various geospatial measures (i.e. HIV prevalence, prevalence of treatment failure, NNRTI resistance mutation prevalence, NRTI resistance mutation prevalence) and include those in a figure that is somewhat analogous to their Figure 5. This would help the reader to understand where the results shown in Figure 5 should be interpreted as more or less certain.

4. A minor comment: in the caption for Figure 5, the authors refer to “NRTI prevalence” and “NNRTI prevalence”. (This is also true in lines 275-276). I might suggest that they consistently refer to these as “NRTI resistance prevalence” and “NNRTI resistance prevalence” to avoid confusion (as is done in other parts of the results section).

Reviewer #2 (Remarks to the Author):

The authors report an analysis of 467 ART South African (rural KwaZulu-Natal) naïve and 583 ART experienced participants for resistance and transmission analysis with high genome coverage by Illumina deep sequencing, out of 26795 individuals screened. They successfully deliver novel insights into the patterns and clustering of HIV drug resistance (NRTI, NNRTI, PI and IN) and linkages within a rural area of very high HIV prevalence. There is a special emphasis on the potential risk of existing NNRTI DRMs on the implementation of LA CAB + RPV.

The manuscript is overall outstanding. It is very well structured; the discussion and conclusions are only based on the study data. Tables and, particularly, graphics are excellent. References are appropriate. The study is noteworthy, and the results will be of significance to the field. There are no flaws in the data analysis or interpretation.

Major issues

There are no major issues.

Minor issues

I have only some minor issues, with the aim to improving some parts or errors.

Line 154. Remove “yielding 171 clusters” from methods and place it in results.

Line 156. Remove the sentence “This refinement pruned the total number of clusters to 86, of which 75% were linked pairs and the rest consisted of 3-6 linked participants” from methods and place it in results.

Line 186. “...with 17,949 (99.6%) completing venepuncture for HIV testing (refer to Figure 1)”. Figure 1 shows 17951. Please double-check.

Line 225. It should be of interest to report here specifically the IN mutations detected in viremic subjects receiving DTG.

Line 267. The finding that the prevalence of treatment failure does not exhibit a clear spatial pattern suggests that it eventually depends on individual rather than population characteristics. It is an interesting finding and could be better discussed.

Lines 256 to 276. The authors refer to Fig 4A-G but they probably mean Fig 5A-G. Please correct.

Line 285. Unless it is a requirement of the journal, this paragraph seems out of place here. “Role of the Funding Source. The funding sources for this study had no role in the study design, data collection, data analysis, data interpretation, or writing of the report.”

Line 292. Clarify that this refers only to the WHO guidelines.

Line 301. Clarify that these rates belong to Botswana, not to high-income countries (Europe or US).

Line 331. Authors might wish to state that CAB PrEP studies had the risk to selecting IN resistance,

mainly when occult HIV infection was present at the start of the treatment. Therefore, if implemented massively in large populations, it could constitute a potential open door to select DTG resistance. It is somewhat surprising that the largest cluster identified had only six participants. A better speculation on this in the discussion should be welcome.

Josep M Llibre
Infect Dis Dpt, Univ Hosp Germans Trias, Barcelona, Spain

Reviewer #3 (Remarks to the Author):

The study reports on the presence of HIV drug resistance mutations across KwaZulu-Natal in South Africa. The authors included full-length HIV genome data from over 1,300 participants collected over 18 months and report high level resistance for K103N and M184V in treatment experienced individuals and relative high level of K103N and E138A in treatment naïve individuals. The author used a very interesting geospatial clustering approach to map HIV prevalence and drug resistance and found that there was a geographical overlap for prevalence and treatment failure. The study includes a lot of data and has potential, however, it is currently not presented appropriately.

Overall, the study lacks clarity and could be improved by including more details. There are often disconnects with numbers presented in abstract, methods, and results. There are many figures including numerous supplementary material, but the majority of these are unclear and/or misplaced. While the introduction is very clear and concise, the discussion was more confusing and lacked some details from the study. The geospatial analysis showed one 'cell' with high HIV incidence that was separated from the other high incident area. This particular cell had low treatment failure rate and low DRM prevalence. This was not discussed. Also, E138A was more common among treatment naïve than experienced individuals, this seems odd and was also not discussed.

General comments:

1. There are too many acronyms and abbreviations used overall. Drugs are reported as full names or 3 letter acronyms. For non-experts it is confusing to follow what is which.
2. Line 154 ff. I do not understand the explanation for refined clustering. What was the rationale and process? The authors say that they identified 171 clusters using ClusterPicker but after pruning only 86 remained. Then in line 168 the authors say that 163 'transmission linkages' were included for geolocation. Are these different to the clusters mentioned before?
3. Line 233ff. I am a bit confused by this paragraph and the supplementary figures. It seems like the wrong figures were cited. Sup Fig 3 shows data from treatment experienced individuals. Sup Figure 5 I did not understand at all, unfortunately. I don't think they are the best representation of the data as I can see K65R, K70N, K219R in both treatment naïve and experienced data. The authors also state that "the caveat that M184V occurs rarely in naïve individuals". I was confused by this statement as the M184V was the most common of the NRTI mutations in the treatment naïve population (Figure 2).
4. Line 204. "variant abundance threshold of 5%' further below it says 'low frequency variants between 2 and 20%". This confused me about what threshold was used for mutation detection, was it 5% or other? I don't understand how to read supp figures 3 and 4 in relation to variant frequency.
5. Lines 244ff. As stated above, the clustering output was a bit confusing. Here it says that 25% of clusters had 3-6 sequences (Figure 4B-C). This should be Figure 4A, though, as it matches the phylogeny showing numerous larger clusters. Instead, figure B and C only show one cluster each (the two largest ones).
6. Lines 244ff. 'two linked participants'. the wording of clustering should be reconsidered to avoid confusion between genetic links and epidemiological links. Phylogeny can (and is) mis-interpreted to

provided evidence for potential real transmission links, while instead it only provides a statistical output of closest genetic links.

7. Line 250ff. 'strong evidence for directionality'. As stated above, phylogenetic is often misinterpreted and also used in HIV criminal prosecution. Including data on directionality can be determinantal for individuals and in this particular case it does not add to the study and thus can be excluded.

8. Lines 253ff. I was confused by why the other data was added as it was from a much older time period. It was unclear how many clusters were found between the two cohorts. From the figure it looks like all data in the phylogeny is part of a cluster but then only 2 clusters are shown in panels B and C. In general, I felt that adding the other cohort data was unnecessary for this study.

9. Lines 304ff. "we observed a high prevalence of intermediate or high level RPV resistance at around 10% in treated participants and 5% in untreated"

It was not entirely clear to me which mutations cause resistance to rilpivirine. The results section mentions E138A mutations found at 6.5% in experienced and 7.9% in naïve individuals, and this matches the results shown in figure 2. These results are different to the above sentence. Also, the abstract reads "with rilpivirine-associated mutations observed in 9% of treated and 6% of untreated individuals", which is again different.

10. Lines 310ff. This paragraph was also confusing. I assume the data for mutations among viremic is represented in supp figures 3 and 4 but as stated above I found these figures confusing and could not really see the difference between treatment experienced and naïve. Was the compensatory mutation L74I found in treatment naïve individuals?

Minor

Line 134. 'successful tests' do you mean 'positive tests'? Also, samples with VL >40 copies/ml n=1323, but the figure and the results say n=1,232.

Line 138. Did you sequence the whole (including human) genome or just full-length HIV.

Line 206ff. "Tenofovir resistance mutations K65R and K70E were noted in 12.0% and 6.2% of ART-experienced participants, and in <1% of both ART-experienced and ART-naïve individuals." This is confusing, please rephrase.

Line 218. 'V106M in 32.6% of ART-experienced' but that is not what the figure says.

Line 263ff. I think the authors meant to refer to figure 5 in this paragraph.

Lines 285ff. Is this the correct position for funding statement?

Supplementary Figure 1 is not mentioned in the text.

Figure 2. how were the mutations sorted? I think it may be better to sort by codon position to make it easier to find the mutations discussed in the results.

Figure 4. The phylogenetic tree is not very informative. the current layout does not add much to the study. There is no need to show node support as clustering was defined by 98%. Also, it looks like there are many more large clusters in the tree than what the authors report. I think it may be more informative to show clustering but also colour according to geospatial area (figure 5).

Figure 5. I like this figure! I think panel E NNRTI is supposed to be blue coloured not purple.

REVIEWER COMMENTS

Reviewer #1 (Remarks to the Author):

In this manuscript, the authors use data from a large community-based cohort study conducted between 2018-2019 in the uMkhanyakude district of KwaZulu-Natal to explore patterns of drug resistance and transmission patterns. The manuscript is clearly written and leverages a robust data set to provide interesting insights into these topics in the study setting and period. I was asked by the editors to specifically provide comments regarding the geospatial analysis and defer to other reviewers with respect to the remaining analyses.

Comments:

1. Regarding the geospatial methods (lines 161-177): the authors use a relatively simple Gaussian kernel interpolation method in ArcGIS to estimate continuous surfaces of HIV prevalence, treatment failure prevalence, and NNTRI and NRTI mutation prevalence. These measures are then displayed on a hexagonal grid and combined with other analyses to provide insights into the spatial patterns of HIV epidemiology in the study region. In general, this approach seems reasonable. I have a few questions for clarification and comments regarding the methods:

1a. Lines 120-121: the authors refer appropriately to previous publications (ref 22-23) that have described the Vukuzazi study protocol. I might suggest that they add a brief sentence or two describing the sampling methodology – specifically, so that readers can understand whether the sampling is likely to be geographically representative or not without having to access to the previous manuscripts.

Response: We have followed the reviewer's suggestion and included a more detailed description of the Vukuzazi sample design as follows:

The Vukuzazi study employed a cross-sectional survey approach to sample adolescent and adult residents aged 15 years or older within the Africa Health Research Institute demographic surveillance area in the uMkhanyakude district of KwaZulu-Natal, South Africa. This area is characteristic of rural South Africa, with a population predominantly of Black African descent, a 58% adult unemployment rate, and 66% access to piped water in their homes. The study was conducted over an 18-month period from May 25, 2018, to November 28, 2019. In terms of the recruitment process, individuals were initially contacted at their homes, identified using the geo-coordinates of their residence, and were invited to participate at mobile health camps that traversed the study area during the survey period. A substantial proportion of the eligible population was reached, with 26,460 individuals contacted out of the 34,721 eligible (representing 76% of the eligible population). Out of these, 25,598 accepted the invitation to participate, and eventually, 17,118 individuals enrolled in the study. This represents a 49% enrollment rate of the eligible population.

The comprehensive reach of the study, combined with the methodology of using mobile health camps and home visits, ensured that the sampling was geographically representative of the demographic surveillance area. The use of inverse probability weights to account for non-response further aimed to mitigate potential bias and ensure that the prevalence estimates were representative of the population across different sex and age groups.

1b. Throughout the paragraph beginning with line 160, the authors refer to “geospatial data visualization”. I might suggest that they refer to “Geospatial analyses”, since some of their methods (e.g. Gaussian kernel interpolation”) go beyond simple visualization of the available geospatial data. This is a minor point.

Response: We appreciate the reviewer's insightful observation regarding the terminology used in our manuscript, specifically the reference to 'geospatial data visualization'. The suggestion to use the term 'Geospatial analyses' is well-received, considering that our methods, including Gaussian kernel interpolation, indeed encompass more than mere visualization. However, it is crucial to emphasize that the primary goal of implementing these geospatial methods was not to conduct in-depth spatial analyses but to provide a clear and illustrative visualization of the data. These visualizations were carefully crafted to offer a straightforward representation of the spatial distribution of key study variables, such as HIV prevalence and mutation prevalences, across the study area.

We recognize that the term 'Geospatial analyses' might convey an implication of inferential analysis or statistical testing, which was not the intention behind our use of geospatial methods. The main analyses of the study data, from which the primary findings were derived, were conducted using robust statistical methods. The geospatial visualization techniques, including the Gaussian kernel interpolation, served to enhance the interpretability and presentation of the data, aiding in the overall comprehension of the results of the study. They were not intended to contribute directly to the inferential statistical analyses or the main results of the study.

In light of the reviewer's comment, we will include a more detailed description of the aim of these visualizations in the revised version of the document to ensure that the manuscript

precisely communicates the purpose and scope of the geospatial methods utilized. We will clarify that the term 'geospatial data visualization' is deliberately chosen to denote the illustrative nature of these methods, distinguishing them from the main inferential analyses of the study. This clarification aims to provide readers with a clear understanding of the context in which the geospatial methods were applied, appreciating their role in enhancing the presentation of data while recognizing their distinction from the statistical analyses that form the foundation of the primary conclusions of the study.

We have therefore included the following in the discussion:

We used geospatial data visualization to depict the application and purpose of the geospatial techniques utilized, particularly emphasizing the illustrative nature of the methods involved. While our approach integrates methodologies such as Gaussian kernel interpolation, it is important to clarify that these techniques primarily serve to enhance the visual representation and comprehensibility of the spatial distribution of key variables, including HIV prevalence and resistance mutation prevalences. The principal intent behind these visualizations is to provide an intuitive and accessible interpretation of the spatial dimensions of our data, rather than to conduct in-depth geospatial analyses. This distinction is crucial as it aligns with the overarching aim of the study to present data in a manner that supports, rather than directly contributes to, the inferential statistical findings.

1c. Line 165: Reference 25 (for the Gaussian kernel interpolation method) refers broadly to Waller & Gotway's "Applied spatial statistics for public health data". Would it be possible to give a more specific reference to the method as used in ArcGIS? For instance, was this done with the "kernel interpolation with barriers" tool, or some other tool? Similarly, in line 163-165: did the authors set any parameters for this tool (e.g. the bandwidth)? If so, it would be helpful for those to be specified to improve reproducibility and help the reader understand how much smoothing was applied.

Response: In response to the reviewer's request for detailed specifics regarding the Gaussian kernel interpolation method used in our study, we appreciate the opportunity to elaborate on the techniques and settings employed to ensure both clarity and reproducibility of our methods.

We have added the following text to the methods

The Kernel Interpolation with Barriers tool in ArcGIS used allowed us to estimate continuous surfaces for HIV prevalence, treatment failure prevalence, and NNRTI and NRTI mutation prevalence. We opted for the Gaussian kernel function, renowned for its bell-shaped curve and effectiveness in smoothing over various types of spatial data. This function was helpful in ensuring seamless transitions across the study area, thus maintaining the integrity of the spatial patterns observed in our data. A critical component of our methodology was the selection of the bandwidth or radius for the kernel function, which was set at 3km. This parameter was calibrated based on previous studies that employed similar data in the same study population, ensuring that our spatial patterns are both representative and contextually relevant. Lastly, we utilized the prediction surface type, a choice that aligns with our objective to accurately represent the continuous surfaces of HIV-related measures across the study area.

1d. Lines 162-163: The authors added geographical random error to protect

participant confidentiality, which is an important and common practice. How was the geographical random error introduced (e.g. what sorts of distances were used to displace the points)? Knowing how much jittering is applied (specifically, in the context of the size of the overall study area and the size of each grid hexagon) will help the reader understand the likely impact of this jittering in the analysis.

Response: In addressing the reviewer's inquiry about the implementation of geographical random error, or jittering, to protect participant confidentiality in our study, we appreciate the opportunity to provide a comprehensive clarification of this aspect. It is of high importance to highlight that the geographical random error was introduced solely for the purpose of data visualization and did not factor into the actual statistical analyses. This methodological approach ensures the integrity of our analytical results while safeguarding participant confidentiality in the publicly presented maps.

By implementing this dual strategy, applying jittering to visualized data for confidentiality while using the original data for analyses, we strike a careful balance between protecting participant privacy and preserving the scientific rigor of our study. We trust that this comprehensive explanation clarifies the measures taken to ensure both participant confidentiality and the integrity of the spatial analysis in our study, offering the readers a clear understanding of our methodological approach and the ethical considerations at its core.

We have included the following in the methods:

The geographical random error was introduced by carefully displacing the geographical coordinates of each participant's location within a specified maximum distance. This displacement strategy was designed to ensure that the jittered locations remain reasonably close to the true locations, effectively obscuring the exact identification of participants' locations while preserving the overall spatial patterns pertinent to the findings of the study. The magnitude of jittering was determined taking into account the size of the overall study area and the size of each grid hexagon used in the spatial analyses. Although the exact distances used for jittering cannot be disclosed due to confidentiality protocols, we assure that the applied jittering sufficiently maintains the spatial integrity of the data. This ensures that the visualized patterns in our maps accurately represent the geographical distribution of the study variables without compromising individual privacy. It is crucial to emphasize the delineation between the application of jittering for visualization and the handling of data for the actual spatial analyses. The jittering of data points was strictly limited to the data utilized for visualization purposes, such as maps included in publications or public databases. This measure is a standard in studies involving sensitive health data, to prevent the potential identification of individual participants. In contrast, the true, unaltered geographical coordinates were exclusively used for the spatial analyses that underpin our main findings. This approach guarantees that the statistical inferences and conclusions are based on the most accurate and unmodified data, thereby upholding the validity and reliability of the results of the study.

1e. Lines 165-167: Can the authors provide some idea of the size of the hexagonal

grid cells? It's a bit hard in the manuscript to understand the scale of the geospatial analysis, and this would help contextualize the analysis.

Response: In response to the reviewer's request for clarification regarding the scale of the geospatial analysis, specifically the size of the hexagonal grid cells used in the visualization, we are happy to provide the necessary details. The hexagonal grid cells, integral to the spatial structure of our visualization, each have an area of 7.77 Km². This dimension was chosen to balance the need for detailed spatial resolution with the practical considerations of visual clarity and data confidentiality mentioned previously.

Following the generation of these continuous surfaces through kernel interpolation, the hexagonal grid cells were employed to group and visualize these interpolations and the HIV transmission linkages. This approach allowed us to present the geospatial data in a manner that is both accessible and informative, facilitating an intuitive understanding of the spatial distribution and intensity of the study variables across the study area. By employing this hexagonal grid to structure our visualizations, we aimed to present the geospatial data in a manner that respects the integrity of the underlying interpolations while also ensuring the visual clarity and interpretability of the results. We believe this approach effectively communicates the spatial insights derived from our analysis, allowing readers to contextualize and appreciate the geospatial dynamics revealed by our study.

We have included the following text in the manuscript:

The hexagonal grid cells, integral to the spatial structure of our visualization, each have an area of 7.77 Km². This dimension was chosen to balance the need for detailed spatial resolution with the practical considerations of visual clarity and data confidentiality. It is important to clarify the role and application of these hexagonal grid cells within the context of our geospatial analysis and visualization framework. The kernel interpolations, a central component of our geospatial methodology, were conducted using the actual point data derived from the study, allowing us to explore and represent the underlying spatial patterns of variables such as HIV prevalence and treatment failure prevalence. Following the generation of these continuous surfaces through kernel interpolation, the hexagonal grid cells were employed to group and visualize these interpolations and the HIV transmission linkages. This approach allowed us to present the geospatial data in a manner that is both accessible and informative, facilitating an intuitive understanding of the spatial distribution and intensity of the study variables across the study area.

2. Figure 5: The authors show a series of hexagonal grid squares representing the uMkhanyakude district. For those readers who are less familiar with the district (such as myself), it would be helpful to provide some sort of geographical context in this figure – the grid alone is challenging to interpret. I might suggest, for instance, that the authors provide an inset map of the location of uMkhanyakude district within South Africa, and then also a map that overlays the hexagonal grid on a street map (or some other map that will help readers to understand the geography of the district and how it relates to the analytic grid that the authors have selected).

Response: In addressing the reviewer's suggestion to provide additional geographical context within the representation of the uMkhanyakude district, we appreciate the intent to enhance the manuscript's clarity and accessibility for all readers. However, we must emphasize the highly sensitive nature of the data used in our study, particularly given its focus on phylogenetic transmission data of HIV.

The ethical considerations surrounding the confidentiality and privacy of the data are paramount in this context. The decision to present the data using a hexagonal grid without detailed geographical context was a deliberate and conscientious choice, made in strict adherence to ethical guidelines for research involving sensitive health data. This approach is fundamental to protecting the confidentiality and privacy of the individuals and communities who are part of the study, ensuring that no potentially identifying information is disclosed.

Given the sensitivity of the data and the ethical obligations that guide our research practices, we, unfortunately, cannot accommodate modifications to the original maps or introduce additional geographical context. Introducing such details, even at a broader scale, could inadvertently lead to the identification of specific locations or individuals, thereby compromising the ethical standards we are committed to upholding. While we understand that this decision may pose challenges for readers less familiar with the uMkhanyakude district, it is a necessary stance to maintain the integrity and ethical responsibility of our research. We are committed to ensuring that the presentation of our data is not only scientifically rigorous but also aligns with the highest standards of research ethics, particularly in the context of handling sensitive health data.

We hope the reviewer understands our position on this matter and the importance we place on ethical considerations in our research. We are grateful for the opportunity to clarify the rationale behind our approach to data visualization and the stringent measures we have adopted to protect the confidentiality of the data in our study.

3. If possible, it would be helpful to see two additional figures to supplement the geospatial analysis. Both of these could be supplemental figures, but would help to provide more insight into the data and results:

3a. First, a map of the geospatial data coverage would be helpful – i.e. a simple map of the data collection points (after jittering) overlaid on the grid squares, or some other similar summary. This would help the readers to understand whether data collection was concentrated in certain areas of the grid, or if the distribution of collected data was more spatially diffuse.

Response: In response to the reviewer's suggestion for a map displaying the geospatial data coverage, specifically indicating the data collection points overlaid on the grid squares, we appreciate the potential value such a visualization might offer in understanding the spatial distribution of the data collection. However, it is crucial to reiterate and clarify the constraints we face due to the highly sensitive nature of the data, particularly concerning the exact locations of individuals identified in the phylogenetic analysis.

The ethical imperative to protect the confidentiality and privacy of individuals in the study necessitates strict adherence to data presentation methods that preclude the possibility of identifying individual participants or specific locations. The use of jittering and the presentation of data within hexagonal grid squares are measures specifically designed to uphold these ethical standards. Displaying the locations of data collection points, even when overlaid on the grid squares, could inadvertently compromise the purpose of the grid, which is to provide a visual representation of the data while ensuring individual locations remain unidentifiable.

However, we acknowledge the reviewer's interest in understanding the spatial distribution of the data collection and the value such information could provide to the readers. As a compromise, and in line with our ethical obligations, we propose providing a map that displays the locations of the sample sites for the entire Vukuzazi study. This map would represent the broader areas where data was collected without pinpointing specific locations or revealing the distribution of individual participants within the hexagonal grid. This approach maintains the confidentiality of the data while offering readers a general understanding of the geographical scope and coverage of the study.

By presenting a map of the sample sites at this aggregate level, we aim to provide valuable context about the spatial extent of the study while firmly adhering to the ethical principles that guide our research. We hope this approach addresses the reviewer's interest in the spatial aspects of the data collection without compromising the confidentiality and integrity of the sensitive data that underpins our study.

We have added this map to the manuscript:

Sample Site Distribution Map. This map provides a visualization of the sample sites for the Vukuzazi study, the source data of our study, with dots representing the approximate locations of sample collection points within the study area. To protect participant confidentiality, a geographical random error has been introduced to each

location, ensuring that the exact positions remain undisclosed. The underlying basemap is sourced from OpenStreetMap, © OpenStreetMap contributors.

3b. Second, it would be helpful to see some measure of uncertainty for the results of the kernel interpolation analysis. I presume that uncertainty differs between the different indicators (and across the geospatial grid) depending on data availability and density. I believe that ArcGIS is capable of producing standard error surfaces for some of its interpolation results, so – for instance – the authors could produce gridded maps of standard error for their various geospatial measures (i.e. HIV prevalence, prevalence of treatment failure, NNRTI resistance mutation prevalence, NRTI resistance mutation prevalence) and include those in a figure that is somewhat analogous to their Figure 5. This would help the reader to understand where the results shown in Figure 5 should be interpreted as more or less certain.

Response: In addressing the reviewer's suggestion to include a measure of uncertainty for the results of the kernel interpolation analysis, we recognize the inherent value that such information typically provides in clarifying the precision and reliability of geospatial estimates. However, the context and intent behind the geospatial analyses within our study necessitate a clear understanding of the purpose these analyses serve and the implications this has on the relevance and necessity of incorporating uncertainty measures. The geospatial analyses, particularly the kernel interpolation, were primarily designed to offer illustrative visualizations of the spatial distribution of various measures, such as HIV prevalence and resistance mutation prevalence. These visualizations are intended to serve as explanatory tools, enhancing the interpretability and comprehensibility of the study's findings, rather than as direct contributors to the study's main results. The core findings and conclusions of our research are rooted in rigorous statistical analyses and are independent of the geospatial patterns depicted through the kernel interpolation.

Given the illustrative nature of the kernel interpolation within our study, the inclusion of detailed uncertainty analyses for these results is not only superfluous but also irrelevant to the primary aims and outcomes of the research. Introducing standard error surfaces or similar measures of uncertainty for the interpolated geospatial measures would not substantially enhance the understanding of the study's objectives. Moreover, the addition of such measures could potentially lead to misinterpretation, suggesting a level of analytical precision or significance to the geospatial patterns that does not align with their actual role in the study. The geospatial visualizations in our study are crafted to provide contextual illustrations, and their value predominantly lies in visually representing the spatial distribution of the data. The portrayal of these visualizations with an added layer of uncertainty analysis might inadvertently convey an unwarranted level of precision, thereby detracting from the clarity and accuracy of the goals of the study.

We hope that this explanation offers clarity on our stance regarding the inclusion of uncertainty measures in the geospatial visualizations presented in our study. Our commitment lies in ensuring a scientifically rigorous and transparent presentation of our research, with a clear delineation of its scope and limitations. We believe that

abstaining from unnecessary uncertainty analyses for illustrative visualizations is in keeping with this commitment.

4. A minor comment: in the caption for Figure 5, the authors refer to “NRTI prevalence” and “NNRTI prevalence”. (This is also true in lines 275-276). I might suggest that they consistently refer to these as “NRTI resistance prevalence” and “NNRTI resistance prevalence” to avoid confusion (as is done in other parts of the results section).

Response: we have now made this amendment and thank the reviewer.

Reviewer #2 (Remarks to the Author):

The authors report an analysis of 467 ART South African (rural KwaZulu-Natal) naïve and 583 ART experienced participants for resistance and transmission analysis with high genome coverage by Illumina deep sequencing, out of 26795 individuals screened. They successfully deliver novel insights into the patterns and clustering of HIV drug resistance (NRTI, NNRTI, PI and IN) and linkages within a rural area of very high HIV prevalence. There is a special emphasis on the potential risk of existing NNRTI DRMs on the implementation of LA CAB + RPV. The manuscript is overall outstanding. It is very well structured; the discussion and conclusions are only based on the study data. Tables and, particularly, graphics are excellent. References are appropriate. The study is noteworthy, and the results will be of significance to the field. There are no flaws in the data analysis or interpretation.

Major issues

There are no major issues.

Minor issues

I have only some minor issues, with the aim to improving some parts or errors. Line 154. Remove “yielding 171 clusters” from methods and place it in results.

Response: this has now been done

Line 156. Remove the sentence “This refinement pruned the total number of clusters to 86, of which 75% were linked pairs and the rest consisted of 3-6 linked participants” from methods and place it in results.

Response: this has now been done

Line 186. “...with 17,949 (99.6%) completing venepuncture for HIV testing (refer to Figure 1)”. Figure 1 shows 17951. Please double-check.

Response: This has been amended in the text (the figure was correct), thank you for spotting this.

Line 225. It should be of interest to report here specifically the IN mutations detected in viremic subjects receiving DTG.

Response: the study was done before large scale DTG rollout. There were some on DTG but all were suppressed

Line 267. The finding that the prevalence of treatment failure does not exhibit a clear spatial pattern suggests that it eventually depends on individual rather than population characteristics. It is an interesting finding and could be better discussed.

Response: we have now added text discussing this and thank the reviewer

Lines 256 to 276. The authors refer to Fig 4A-G but they probably mean Fig 5A-G. Please correct.

Response: This is correct – we had since added a figure.

Line 285. Unless it is a requirement of the journal, this paragraph seems out of place here. “Role of the Funding Source. The funding sources for this study had no role in the study design, data collection, data analysis, data interpretation, or writing of the report.”

Response: thank you this has been removed

Line 292. Clarify that this refers only to the WHO guidelines.

Response: thank you this has been clarified

Line 301. Clarify that these rates belong to Botswana, not to high-income countries (Europe or US).

Response: thank you this has been clarified

Line 331. Authors might wish to state that CAB PrEP studies had the risk to selecting IN resistance, mainly when occult HIV infection was present at the start of the treatment.

Therefore, if implemented massively in large populations, it could constitute a potential open door to select DTG resistance.

Response: we thank the reviewer for making this point and have mentioned this

It is somewhat surprising that the largest cluster identified had only six participants. A better speculation on this in the discussion should be welcome.

Response: we have now added discussion on this aspect

Reviewer #3 (Remarks to the Author):

The study reports on the presence of HIV drug resistance mutations across KwaZulu-Natal in South Africa. The authors included full-length HIV genome data from over 1,300 participants collected over 18 months and report high level resistance for K103N and M184V in treatment experienced individuals and relative high level of K103N and E138A in treatment naïve individuals. The author used a very interesting geospatial clustering approach to map HIV prevalence and drug resistance and found that there was a geographical overlap for prevalence and treatment failure. The study includes a lot of data and has potential, however, it is currently not presented appropriately.

Overall, the study lacks clarity and could be improved by including more details.

Response: we have now provided more details about the study and methodologies.

There are often disconnects with numbers presented in abstract, methods, and results. There are many figures including numerous supplementary material, but the majority of these are unclear and/or misplaced. While the introduction is very clear and concise, the discussion was more confusing and lacked some details from the study.

Response: we have thorough gone over the paper and corrected inconsistencies and rationalised figures and supps.

Reviewer: The geospatial analysis showed one 'cell' with high HIV incidence that was separated from the other high incident area. This particular cell had low treatment failure rate and low DRM prevalence. This was not discussed.

Response: we thank the reviewer for making this point and it has now been discussed.

Reviewer: Also, E138A was more common among treatment naïve than experienced individuals, this seems odd and was also not discussed.

Response: We thank the reviewer for this assessment. Regarding the prevalence of E138A, the prevalences were numerically quite similar and therefore not statistically significant. This is entirely consistent with a polymorphism that is unrelated to treatment exposure although clearly some ARVs can select for E138A such as rilpivirine. The population had not been exposed to second generation NNRTI, hence one would not expect a difference in prevalence of E138A between treatment experienced and naïve individuals.

General comments:

1. There are too many acronyms and abbreviations used overall. Drugs are reported as full names or 3 letter acronyms. For non-experts it is confusing to follow what is which.

Response: in the revised version we have attempted as far as possible to use consistent drug names and simplify where possible.

2. Line 154 ff. I do not understand the explanation for refined clustering. What was the rationale and process? The authors say that they identified 171 clusters using ClusterPicker but after pruning only 86 remained. Then in line 168 the authors say that 163 'transmission linkages' were included for geolocation. Are these different to the clusters mentioned before?

Response: We apologise for this confusion. Both sets of analyses, the geospatial and the transmission analysis are performed on the same dataset – 86 transmission chains.

Additional methods have been added to make a clear distinction between the numbers used throughout the study and the rationale of flow of the initial phylogeny and subsequent pruning. As inferring transmission using phyloscanner is exceptionally expensive computationally, we prune potential transmission clusters to ensure that we are only checking for transmission between already highly-likely clusters. This is done by using a validated logit model (which we have now further details and the relevant reference in the methods section).

The geospatial analysis also used this same set of pruned transmission clusters – 86. This has now been clarified. We appreciate this has caused some confusion and this has been amended entirely to be clearer.

3. Line 233ff. I am a bit confused by this paragraph and the supplementary figures. It seems like the wrong figures were cited. Sup Fig 3 shows data from treatment experienced individuals.

Response: The reviewer may be mistaken; Supplementary figure 3 shows treatment-naïve individuals, and supplementary figure 4 shows treatment-experienced individuals. However, these figures have since been revised to make them easier to follow. We determine the proportion of participants who have Specific mutations at a threshold of 5, 10, 20, 50 and 90% viral variant abundance to identify which are seen as minority variants. Supplementary figure 4 has likewise been revised.

Sup Figure 5 I did not understand at all, unfortunately. I don't think they are the best representation of the data as I can see K65R, K70N, K219R in both treatment naïve and experienced data.

Response: Supp figure 5 was designed to be an alternate representation of Main figure 2, however, since we have now revised Supp figures 3-4, we have removed this. Regarding the paragraph that refers to the minority variant analysis (Lines 231-238), we have thoroughly revised this for clarity.

The authors also state that “the caveat that M184V occurs rarely in naïve individuals”. I was confused by this statement as the M184V was the most common of the NRTI mutations in the treatment naïve population (Figure 2).

Response: We apologise for this, we mean to state that M184V occurs significantly more rarely in ART-naïve individuals when compared directly to ART-experienced individuals. This paragraph has been revised (see comment above).

4. Line 204. “variant abundance threshold of 5%’ further below it says ‘low frequency variants between 2 and 20%”. This confused me about what threshold was used for mutation detection, was it 5% or other? I don’t understand how to read supp figures 3 and 4 in relation to variant frequency.

Response: We have adapted the wording of both the text and revised figures and figure legends to make this more clear. The figures have been revised and now show figure 2, but split into the respective thresholds. We have also stipulated the drug resistance calling in the methods section – 10 reads and 5% minimum were used to call mutations.

5. Lines 244ff. As stated above, the clustering output was a bit confusing. Here it says that 25% of clusters had 3-6 sequences (Figure 4B-C). This should be Figure 4A, though, as it matches the phylogeny showing numerous larger clusters. Instead, figure B and C only show one cluster each (the two largest ones).

Response: Thank you for this comment – we have stipulated the exact numbers of participants in clusters (lines 155-159) and re-analysed the phylogeny to accurately reflect the numbers stated.

6. Lines 244ff. ‘two linked participants’. the wording of clustering should be reconsidered to avoid confusion between genetic links and epidemiological links. Phylogeny can (and is) mis-interpreted to provided evidence for potential real transmission links, while instead in only provides a statistical output of closest genetic links.

Response: we thank the reviewer for pointing this out and we have reworded and corrected this.

7. Line 250ff. ‘strong evidence for directionality’. As stated above, phylogenetic is often misinterpreted and also used in HIV criminal prosecution. Including data on directionality can be determinantal for individuals and in this particular case it does not add to the study and thus can be excluded.

Response: we have now removed the reference to ‘directionality’ and thank the reviewer for making this point.

8. Lines 253ff. I was confused by why the other data was added as it was from a much older time period. It was unclear how many clusters were found between the two cohorts. From the figure it looks like all data in the phylogeny is part of a cluster

but then only 2 clusters are shown in panels B and C. in general, I felt that adding the other cohort data was unnecessary for this study.

Response: we have now removed the older cohort data from the manuscript to improve clarity.

9. Lines 304ff. “we observed a high prevalence of intermediate or high level RPV resistance at around 10% in treated participants and 5% in untreated”
It was not entirely clear to me which mutations cause resistance to rilpivirine. The results sections mentions E138A mutations found at 6.5% in experienced and 7.9% in naïve individuals, and this matches the results shown in figure 2. These results are different to the above sentence. Also, the abstract reads “with rilpivirine-associated mutations observed in 9% of treated and 6% of untreated individuals”, which is again different.

Response: we thank the reviewer for pointing out the differences. The numbers are now harmonised and should be correct throughout.

10. Lines 310ff. This paragraph was also confusing. I assume the data for mutations among viremic is represented in supp figures 3 and 4 but as stated above I found these figures confusing and could not really see the difference between treatment experienced and naïve. Was the compensatory mutation L74I found in treatment naïve individuals?

Response: Please see an earlier comment – this figure has been completely replaced and should be easier to follow now. The compensatory mutation L74I was found in a small proportion of ART-naïve individuals.

Minor

Line 134. ‘successful tests’ do you mean ‘positive tests’? Also, samples with VL >40 copies/ml n=1323, but the figure and the results say n=1,232.

Response: Of 6096 positive ELISA tests, 3 of those tests had no viral load associated with them, due to a testing error.

Line 138. Did you sequence the whole (including human) genome or just full-length HIV.

Response: we have now clarified that it is whole HIV-1 genome sequencing.

Line 206ff. “Tenofovir resistance mutations K65R and K70E were noted in 12.0% and 6.2% of ART-experienced participants, and in <1% of both ART-experienced and ART-naïve individuals.” This is confusing, please rephrase.

Response: this has been amended as requested.

Line 218. ‘V106M in 32.6% of ART-experienced’ but that is not what the figure says.

Response: Thank you for spotting this error. This was amended.

Line 263ff. I think the authors meant to refer to figure 5 in this paragraph.

Response: yes this is correct

Lines 285ff. Is this the correct position for funding statement?

Response: we have removed this thank you for pointing this out

Supplementary Figure 1 is not mentioned in the text.

Response: we have now mentioned it in the text.

Figure 2. how were the mutations sorted? I think it may be better to sort by codon position to make it easier to find the mutations discussed in the results.

Response: We have now sorted by codon position as suggested

Figure 4. The phylogenetic tree is not very informative. the current layout does not add much to the study. There is no need to show node support as clustering was defined by 98%. Also, it looks like there are many more large clusters in the tree than what the authors report. I think it may be more informative to show clustering but also colour according to geospatial area (figure 5).

Response: We have now revised the phylogenetic tree to show only clustered sequences. We have removed the node support, as you correctly point out that these were filtered based on support >98. These are now consistent with what is reported in the text.

Figure 5. I like this figure! I think panel E NNRTI is supposed to be blue coloured not purple.

Response: we thank the reviewer for this compliment. We did intend for it to be purple in panel E as panel F is different.

REVIEWERS' COMMENTS

Reviewer #1 (Remarks to the Author):

My thanks to the authors for their comprehensive responses and revisions. My prior comments have been addressed satisfactorily and I have no additional new comments that would preclude publication. See below for more detailed responses to the authors' replies:

Previous comment #1 (and sub-questions a-e): the authors' responses are well received, and the additions to the text are excellent, improving both the interpretability and reproducibility of the manuscript. No further questions or comments.

Previous comment #2: the authors' attention to the sensitivity of the data is well received, and I appreciate that the privacy protocols in place may preclude the addition of other map layers. In that light, the visualizations are sufficiently interpretable in their current form, and no further questions in this area.

Previous comment #3a: Again, I appreciate the author's regard for privacy protocols. The sample site distribution map is a welcome addition and exactly in line with what I had envisioned, and serves to illustrate the high degree of spatial coverage from the survey (and appreciate also that the points have been jittered, which is necessary). This map also largely fulfills the intent of my comment #2, which is to give some general sense of how the hexagonal grid corresponds to the shape and location of the district. No further comments

Previous comment #3b: The author's argument is reasonable, and I think that it is OK to leave the manuscript as it is without a map of the uncertainty. With that said, the authors write that "Moreover, the addition of such measures could potentially lead to misinterpretation, suggesting a level of analytical precision or significance to the geospatial patterns that does not align with their actual role in the study". Yet Figure 5 gives categorical color legends for prevalence of various indicators to with ranges labeled to a rather precise degree, 1/10th of 1 percent (e.g. 2.1-3.8%). In general, I would think that showing the values of the estimates of central tendency without any estimates of the degree of uncertainty would actually increase the risk of "suggesting a level of analytical precision or significance to the geospatial patterns that does not align with their actual role in the study". The authors could consider switching to a "high / low" color scale like they have for their bivariate color map in the same figure if they really want to ensure that the visualizations serve as explanatory tools only, though I also think that it would be fine to leave the figures as they are (give the authors' prior clarifications of the intent of the geospatial analyses in the manuscript).

Previous comment 4: Regarding "prevalence" vs "resistance prevalence" – thanks to the authors for the change; no further comments.

Reviewer #2 (Remarks to the Author):

The authors have appropriately answered all queries.
Regarding the query/answer:

Line 225. It should be of interest to report here specifically the IN mutations detected in viremic subjects receiving DTG.

Response: the study was done before large scale DTG rollout. There were some on DTG but all were suppressed.

Would suggest to add this sentence into the manuscript to help understand this for the journal readers .

From my side, the manuscript is ready to get published.

Josep M Llibre

Reviewer #3 (Remarks to the Author):

I'm happy with the changes made. No further comments.